# An Overview of EGFR Mechanisms and Their Implications in Targeted Therapies for Glioblastoma

**DOI:** 10.3390/ijms241311110

**Published:** 2023-07-05

**Authors:** Silvia Mara Baez Rodriguez, Amira Kamel, Gheorghe Vasile Ciubotaru, Gelu Onose, Ani-Simona Sevastre, Veronica Sfredel, Suzana Danoiu, Anica Dricu, Ligia Gabriela Tataranu

**Affiliations:** 1Neurosurgical Department, Clinical Emergency Hospital “Bagdasar-Arseni”, Soseaua Berceni 12, 041915 Bucharest, Romania; mara.silvia@icloud.com (S.M.B.R.); kamel.amyra@yahoo.com (A.K.); dr_vghciubotaru@yahoo.com (G.V.C.); ligia.tataranu@umfcd.ro (L.G.T.); 2Neuromuscular Rehabilitation Department, Clinical Emergency Hospital “Bagdasar-Arseni”, Soseaua Berceni 12, 041915 Bucharest, Romania; gelu.onose@umfcd.ro; 3Department of Pharmaceutical Technology, Faculty of Pharmacy, University of Medicine and Pharmacy of Craiova, Str. Petru Rares nr. 2–4, 710204 Craiova, Romania; ani.sevastre@umfcv.ro; 4Department of Physiology, Faculty of Medicine, University of Medicine and Pharmacy of Craiova, Str. Petru Rares nr. 2–4, 710204 Craiova, Romania; veronicasfredel@yahoo.com (V.S.); suzdanoiu@gmail.com (S.D.); 5Department of Biochemistry, Faculty of Medicine, University of Medicine and Pharmacy of Craiova, Str. Petru Rares nr. 2–4, 710204 Craiova, Romania; 6Department of Neurosurgery, Faculty of Medicine, University of Medicine and Pharmacy “Carol Davila”, 020022 Bucharest, Romania

**Keywords:** glioblastoma, RTK, growth factor, EGFR, signaling pathways, EGFR targeted therapy

## Abstract

Despite all of the progress in understanding its molecular biology and pathogenesis, glioblastoma (GBM) is one of the most aggressive types of cancers, and without an efficient treatment modality at the moment, it remains largely incurable. Nowadays, one of the most frequently studied molecules with important implications in the pathogenesis of the classical subtype of GBM is the epidermal growth factor receptor (EGFR). Although many clinical trials aiming to study EGFR targeted therapies have been performed, none of them have reported promising clinical results when used in glioma patients. The resistance of GBM to these therapies was proven to be both acquired and innate, and it seems to be influenced by a cumulus of factors such as ineffective blood–brain barrier penetration, mutations, heterogeneity and compensatory signaling pathways. Recently, it was shown that EGFR possesses kinase-independent (KID) pro-survival functions in cancer cells. It seems imperative to understand how the EGFR signaling pathways function and how they interconnect with other pathways. Furthermore, it is important to identify the mechanisms of drug resistance and to develop better tailored therapeutic agents.

## 1. Introduction

Glioblastoma multiforme (GBM, World Health Organization [WHO] grade 4 glioma) represents the most common, aggressive, and malignant central nervous system (CNS) tumor. Despite advances made in the molecular characterization of GBM and newly available targeted therapeutic approaches, this form of brain cancer has a very poor prognosis with a median survival time estimated at 15 months after maximal treatment [1]. The standard care for glioblastoma includes radical surgery followed by radiotherapy with concomitant chemotherapy with temozolomide (TMZ), an alkylating agent that causes apoptosis generating single and double strand breaks in DNA [2]. Gliomas appear when glial cells—supportive cells of the CNS—become malignant [3,4,5,6,7,8]. Malignant gliomas are highly heterogeneous tumors comprised of multiple populations of cells in different phases of differentiation.

Grade 4 gliomas are characterized by necrosis, vascular proliferation, mitoses, nuclear and cellular atypia [9], and pseudopalisading features [10]. The management of these tumors presents many challenges: their location often limits the surgical accessibility, the blood–brain barrier is almost impossible to penetrate by conventional medical therapies, the infiltrative tumor cells migrate into normal brain parenchyma, the mass effect and also the treatment of these tumors can cause important functional disability, high resistance to radio- and chemotherapy, faulty well-developed neovasculature, the complex and dynamic genome of the tumors, and their molecular and morphological heterogeneity [11,12].

The extent of tumor aggressiveness is determined by several pathological attributes, such as nuclear atypia, necrosis, microvascular proliferation, and intratumoral hemorrhages [13,14]. The microenvironment poses many challenges for the glioma cells, including hypoxia, acidity, and the lack of nutrients; therefore, they modulate their metabolic activity in order to maintain the rapid growth [15,16]. At a microscopic level, GBMs invade the brain beyond the gross radiographic margins of the tumor [17], making it impossible to perform a complete surgical resection.

Until recently, the classification of gliomas was made based only on histological and immunohistochemical criteria. Alongside classical diagnostic methods like functional magnetic resonance (MRI), positron emission tomography, and computed tomography, for GBMs, it is also useful to perform a liquid biopsy—a non-invasive technique which confirms the diagnosis and helps to establish the correct treatment [18,19]—and an additional molecular diagnosis, which helps to provide a more personalized prognosis and enhances the efficacy of therapy [20]. The fifth edition of the WHO Classification of Tumors of the CNS, published in 2021 [21], included the molecular classification of the tumors. This classification describes three genetic criteria for the diagnosis of GBM, IDH-wildtype: telomerase reverse transcriptase (TERT) promoter mutation, epidermal growth factor receptor (EGFR) amplification, and the combined gain of the entire chromosome 7 and loss of entire chromosome 10 [21].

GBMs have been demonstrated to possess intratumoral [22] and intertumoral [23] heterogeneity together with a temporal evolution of their genome [24]. They also present an abundance of cell signaling networks supporting tumor growth and, further, the tumor’s resistance to therapy. The GBM genomic changes were first analyzed in the Cancer Genome Atlas. GBM was classified into four subtypes based on the analysis of mutational changes in 601 genes:Proneural—most common in younger patients. It presents an oligodendrocytic lineage associated with secondary GBM, and enhancing mutations in tumor protein 53 (TP53) and IDH1 genes;Neural—appears in older patients. Derived from astrocytes and oligodendrocytes, it expresses neuron-related genes and no specific mutations;Classical—with no TP53 mutations and enhancing expression of EGFR;Mesenchymal—presents an astroglial lineage, with mutations in neurofibromin 1, phosphatase, and tensin homolog (PTEN) and TP53 genes.

Evidence shows that almost every GBM tumor presents attributes of various subtypes, indicating high levels of intra- and intertumoral heterogeneity [23,25,26] with individual cell variations in their gene expression patterns [27].

A major part of modulating the invasion in GBMs is played by the members of the protein tyrosine kinase family (PTK) [28,29]. The human protein kinase genome encloses 518 protein kinase genes, some of them being responsible for encoding transmembrane receptor tyrosine kinases (RTKs) [30]. RTKs consist of an extracellular ligand-binding domain, a single transmembrane helix domain, and an intracellular region comprised of a juxtamembrane regulatory region, a tyrosine kinase domain (TKD), and a carboxyl (C-) terminal tail [31]. More than 58 mammalian RTKs have been identified [30,32]. Tyrosine phosphorylation plays an important role in signal transduction, modulating a large series of cellular events under both physiological as well as pathological conditions, resulting in cell differentiation, proliferation, migration, and survival [33,34]. The activation of receptor tyrosine kinases (RTKs) results in their oligomerization, a process that occurs when specific ligands bind to their extracellular domain, leading to the activation of the intracellular domain. Then, with the help of recruitment proteins, it initiates a signaling cascade, with numerous specific signaling pathways determining particular cellular responses depending on the cell type and the activated signal transduction pathway [35].

The RTK dimerization can take place in four ways:completely ligand mediated, with no direct contact between the extracellular regions of the two receptors [36];completely receptor mediated, with no physical interaction between two activating ligands—as in the case of EGFR [37];ligand homodimers attach themselves to two receptor molecules and then interact across the dimer interface [38];a combination of direct receptor–receptor contacts and bivalent ligand binding, with the cooperation of accessory molecules in the receptor dimerization [39,40].

A portion of the RTKs can form dimers or high-order oligomers without the participation of activating ligands, with monomers and dimers in a dynamic equilibrium. In the case of EGFR, the monomers are predominant before the ligand-binding phase [41]. The predetermined dimers can be active or inactive, both forms being in dynamic equilibrium with each other. Ligand-induced dimerization occurs after ligand binding, a process that will help to stabilize the active dimer or to activate the inactive one [41,42,43].

The physiological information will be transmitted from the exterior of the cells to the interior by a cascade of processes. First, intra-molecular interactions, that are different for all the receptors, participate in maintaining the TKD in a cis-autoinhibition state [44,45]. After the ligand induces dimerization, this cis-autoinhibition is cleared and the transphosphorylation of key tyrosine residues is induced, leading to damage of the autoinhibitory interactions, with an active conformation of the kinase the end of this process.

As the activation and the autophosphorylation of RTKs occurs, many downstream signaling proteins are recruited. The majority of autophosphorylation sites work as binding sites for Src homology-2 (SH2) and phosphotyrosine-binding (PTB) domain. Both domains contain signaling proteins. The SH2 domain can be recruited either to the receptor directly or indirectly with the participation of docking proteins that use their PTB domains to bind to the RTKs. Additional signaling molecules that contain SH2 or other domains are recruited by the docking proteins [33,46]. The activated RTKs may regulate various signaling pathways with the participation of several docking proteins. Some of these pathways are represented by RAS/MAPK, PI-3 K/AKT, and JAK2/STAT [47].

During glioma initiation and progression, aberrant RTK activation may occur. The abnormal activation of RTK in cancers could be the result of four mechanisms: gain-of-function mutation, genomic amplification, chromosomal rearrangements, and/or autocrine activation. Evidence shows that these tumor-initiating cascades, through signaling cross-talks, may be responsible for the malignant transformation of cells, resistance to treatment, and relapse.

The most amplified RTK in GBM is the epidermal growth factor receptor (EGFR) [48]. EGFR gene amplification is detected in 57.4% patients presenting primary GBM [23] specific to the classical subtype of glioma [25]. It results in high levels of EGFR protein, promoting tumorigenesis and progression [23].

In recent years, clinical trials testing new drugs and therapies have been conducted, but targeted therapies against EGFR have not yet shown any clinical benefit. Ineffective blood–brain barrier penetration, heterogeneity, the presence of mutations, and the compensatory signaling pathways may play a significative role in the drug resistance. A deeper understanding of the EGFR signaling networks and their interrelations with other pathways is essential to determine the mechanisms of resistance and to develop more optimized and effective therapeutic strategies.

In this review, we present the process of the activation of EGFRs under physiological conditions and several mechanisms of aberrant activation in glioblastomas that present important implications for the development of anti-cancer therapies.

## 2. EGFR Biology

Epidermal growth factor belongs to the large family of RTKs. The EGFR family contains four members: designated EGFR (known as ErbB1/HER1) located on chromosome 7p12, ErbB2 (HER2/Neu), ErbB3 (HER3), and ErbB4 (HER4) [49]. EGFR has the same structure as all RTKs: an extracellular domain (ECD), a single transmembrane domain (TMD), an intracellular juxtamembrane domain (JMD), a tyrosine kinase, and a C-terminal end.

EGFR is activated by two large groups of ligands: the activators, also called the EGF agonists, and the neuregulins [50,51]. More than 40 ligands are involved in signaling control: high-affinity ligands like EGF, transforming growth factor alpha (TGF-α), heparin-binding EGF-like growth factor (HB-EGF), betacellulin, and low-affinity ligands like amphiregulin and epiregulin [52,53]. The ECD of EGFR is formed by two homologous domains, I and III, where ligands bind, and two domains, II and IV, that are rich in cysteine [54]. After the binding of extracellular ligands like reactive astrocytes and microglias secreted by glioma cells and tumor microenvironmental cells [55], EGFR suffers dimerization followed by trans-autophosphorylation of its intracellular domain. This may result an active homodimer when EGFR pairs with another EGFR or a heterodimer when it binds to another member of the EGFR family [56] (Figure 1).

Studies have shown that EGFR has pro-survival kinase-independent functions in cancerous cells, offering a new perspective on EGFR implications in malignancies and targeted cancer therapies [57,58,59].

Furthermore, EGFR was proven to influence the glutamine metabolism in glioma cells [60]. Glutamate, obtained from glutamine by glutaminase conversion, is indispensable for tumor cells because of its role as a forerunner to produce α-KG, which further generates carbon and nitrogen in the tumor cells. Indeed, the study performed by Yang et al. demonstrated that the EGFR signaling pathway activated phosphorylated ELK1, which further induced the GDH1 transcription, increasing the glutamine metabolism [61]. Therefore, EGFR plays a key role in the development and progression of glioma due to its involvement in regulating the proto-oncogene MYC family and in promoting the glutamine metabolism via ELK1 activation.

## 3. EGFR Pathway Activation

The EGFR pathway is activated by more than one mechanism, such as increased ligand production or overexpression, receptor mutation, or a deficient inactivation.

Under normal physiologic conditions, the expression of EGFR is between 4 and 10 x104 receptors per cell [62]. Stimulating the EGFR-specific transcription factor (ETF) will result in higher EGFR RNA expression. The epidermal growth factor, alongside other proteins like E1A, Sp1, and AP2, is capable of regulating the expression of EGFR [58].

Multiple pathways are involved in receptor signaling. EGFR can activate a variety of signal transduction pathways at the same time. One of the most frequently studied pathways is the recruitment and activation of the phosphatidylinositol—3- kinase (PI3K) signaling network, which is responsible for tumor growth caused by the downstream activation of AKT and mTOR proteins [63]. PTEN plays a part in the negative regulation of the pathway. Other important pathways are the RAS/MAPK pathway and the JAK2/STAT pathway [64]. The genomic alterations of EGFR contribute to receptor activation and signaling increase.

### 3.1. Activation of the Extracellular Domain

At the level of EGFR, the dimerization is mediated by receptors. There is no physical contact between the activating ligands. In the inactive state, the receptors’ extracellular domain has a tethered configuration, with the dimerization arm being blocked by intra-molecular links and the tyrosine kinase domain being in an inert state. After ligand binding, the dimerization arm is exposed, followed by the dimerization of the extracellular domain. At the intracellular domain level, the kinase activation is facilitated by the configuration changes [65].

### 3.2. Activation of the Intracellular Domains

After the ligand-induced dimerization, the cis-autoinhibition is discharged—the kinase activity of EGFR being activated—by creating a physical contact between the C-terminal tail of the activator and the N-terminal tail of the receiver. This process results in the trans-phosphorylation of the C-terminal domain of the activator and structural changes to the N-lobe of the receiver [66].

### 3.3. Downstream Signaling

The activation and autophosphorylation of EGFR are followed by the recruitment of downstream signaling proteins such as Src Homology 2 (SH2) and phosphotyrosine binding (PTB) signaling proteins. The SH2 proteins are capable of binding either directly to the receptor or indirectly with the help of docking proteins using PTB domains [46]. Many signaling pathways are then recruited and regulated by EGFR [34,48].

The PI-3K/AKT signaling pathway regulates apoptosis and cell survival. Following the activation and the phosphorylation of EGFR, PI3K is recruited and transported to the cell membrane where it phosphorylates phosphatidylinositol 4,5-biphosphate (PIP2), resulting phosphatidylinositol (3,4,5)-triphosphate (PIP3). Then, PIP3 reacts with AKT and can be either phosphorylated by phosphoinosite-dependent protein kinase-1 (PDK1) at Threonin308, or by the mammalian target of rapamycin complex 2 (mTORC2) at Serine 473. The PI3K/AKT pathway is negatively regulated by the phosphatase and tensin homolog (PTEN). PTEN has the ability to dephosphorylate PIP3 and to remove it from the cellular membrane. This process results in the relocation of AKT in the cytoplasm, with the loss of the ability of being reactivated [67,68].

After the dimerization with human HER3, or through the docking protein GRB2-associated binder 1 (GAB1), activated EGFR is associated with regulatory p85, absolving p85 of its inhibitory effect [69]. GAB1 is a scaffolding protein that is implicated in the attraction of other signaling proteins like PI3K, SHP2, and p120RasGap. It plays an important part in the association of EGFR with the PI3K/Akt signaling pathway, being implicated in others EGFR signaling outputs as well [70,71]. In the future, GAB1 may be an important potential therapeutic target due to its significant role in different types of cancers [72].

The RAS/MAPK signaling pathway requires growth-factor-receptor bound-2 (GRB2) to form a complex with Son of Sevenless (SOS), which is a guanine-nucleotide exchange factor (GEF), and the activation of the RAS G-protein, resulting in the conversion of guanosine diphosphate (GDP) into guanosine triphosphate (GTP) [73]. After these processes, a downstream signaling cascade is activated by RAS and MAPK, resulting in the phosphorylation of the nuclear protein Jun, which, by binding to several nuclear proteins, leads to the formation of the key transcription factor activator protein (AP-1). This protein is responsible for the translation and transcription of the proteins that are responsible for cellular growth and division. GTPase activating proteins (GAPs), such as tumor suppressor neurofibromin 1 (NF1), are responsible for the negative regulation of activated RAS [74].

Signal transduction and activator of transcription 3 (STAT3) is activated by the EGFR regulation of interleukin-6 (IL-6) expression, resulting in pSTAT3. This mechanism is involved in the IL-6/Janus kinase (JAK)/STAT3 loop [34,75,76,77].

## 4. Role of EGFR in the Molecular Pathogenesis of Glioblastoma

### 4.1. Oncogenic Activation of RTKs

Normally, the activity of RTKs is managed by various already-described mechanisms and by various molecules, one of them being tyrosine phosphatase [78]. The result of the transforming abilities of RTKs provoked by several mechanisms, is the broken equilibrium between cell growth and proliferation and cell destruction [33,79]. Normal cells can develop oncogenic properties as a result of constitutive activation causing RTK-induced oncogenesis [80]. This type of activation can be the cause of four mechanisms in human cancers [66]:Gain of function mutations: these types of mutations lead to atypical downstream signal transduction. One of these types of mutations are the “driver mutations” that are able to give a selective growth advantage to the cells [81]. It is possible that the further study of these “driver mutations” could help us to understand how the cancer initiates and progresses and to bring new perspectives for targeted treatments. EGFR TKD is encoded by exons 18–24, while the EGFR mutations appear mostly in exons 18–21, close to the ATP binding pocket [82]. The kinase and the downstream signaling are hyperactivated by these mutations, giving them oncogenic properties [82,83,84]. It was demonstrated that patients with tumors that present somatic EGFR TKD mutations are more sensitive to EGFR TKIs [85,86,87,88,89,90,91]. Other types of mutations can appear in the extracellular domain (ECD), transmembrane domain (TMD), or juxtamembrane domain (JMD) of RTKs. In glioblastoma, missense mutations at the level of EGFR ECD were discovered and were associated with higher expression of EGFR protein, which undergoes phosphorylation when not stimulated by ligand [92,93,94]. Patients with EGFR ECD mutations had poor clinical results when under treatment with EGFR TKIs erlotinib and gefitinib [95,96]. The ECD mutations seem to adopt an inactive form, suggesting that they may be more responsive to EGFR targeted therapies that bind to the inactive form of the receptor [97]. It seems that both intra- and extracellular GBM mutations have, as a result, ligand-independent oncogenic activation.Genomic amplification: EGFR is the most commonly overexpressed RTK in GBM [98]. The prevalence of EGFR gene amplification and overexpression is higher in primary GBM than secondary GBM [99]. The consequence of overexpression is a higher local concentration of the receptor, resulting in elevated RTK signaling and overpowered antagonizing regulatory effects [100]. This overexpression is caused by multiple mechanisms, the most important being gene amplification, followed by transcriptional/translational enhancement [101,102], oncogenic viruses [103], degradation of normal regulatory mechanisms such as the loss of phosphatases [104], and other negative regulators. Gene amplification leads to an increase in the copy number of a specific region of the genome [105] in the form of extrachromosomal elements (double minutes) that are usually high-level amplifications with more than 25 copies, or repeated units at a single locus or throughout the genome (distributed insertions) characterized by low-level amplification with 5 to 25 copies [98,106]. These amplifications can be determined by flaws in the DNA replication, fragile sites at the chromosomal level, or telomere dysfunction [105]. The amplification pattern is quite different in the same tumor type [98].Chromosomal rearrangements: studies have shown that the formation of new tyrosine kinase fusion oncoproteins is caused by numerous chromosomal rearrangements [23,107,108]. It may be of significant importance to identify these fusion proteins, as they can be therapeutically targetable with small molecule inhibitors. Some risk factors are thought to participate in the gene fusion events—topoisomerase poisons [109], exposure to ionizing radiation [110,111], and oxidative stress [112]—but the exact way in which these mechanisms function is not yet known. Although a great number of tyrosine kinase fusions have been described, the structure of the fusion oncoproteins is quite similar. The fusion can arise in either the N-terminal or the C-terminal of the RTK, the tyrosine kinase domain being preserved either way. The genomic breakpoint can appear either downstream of the exons that encode the full kinase domain, in which case the ECD, TMD, and JMD are conserved and the resultant fusion protein will behave like a membrane-bound receptor, or upstream thereof, in which case loss of the ECD, TMD, and JMD occurs and the protein that appears as a result will not be membrane bound. The chimeric fusion proteins that appear as a result of the chromosomal rearrangements lead to oncogene addiction [113,114]. The use of target-specific TKIs against RTK fusions may be a good weapon in the battle against numerous types of RTK fusion-driven cancers.Autocrine activation: communication between cells is carried out with the help of messengers, like growth factors and cytokines, released by secretory cells. When the target cells are also the secretory cells, the signaling is called “autocrine” [115]. This type of autocrine activation can lead to tumor formation and clonal expansion [116]. In association with other autocrine growth pathways, the autocrine activation loop of RTKs can lead to tumor formation. The autocrine pathways can be used as a target for cancer therapy [117]. The wild-type EGFR ligands, like TGF-alpha and HB-EGF, are generally increased in glioblastoma, leading to an autocrine loop that results in the growth of glioma cells [118]. GBM expresses EGFRvIII, which does not bind ligands and is thought to be more tumorigenic than wild-type EGFR. TGF-α and HB-EGF induce the expression of EGFRvIII, implying that EGFRvIII may create an autocrine loop with wild-type EGFR, which plays an important role in signal transduction in glioblastoma cells [119].

Another type of RTK activating mechanism is special intragenic partial duplication -kinase domain duplication (KDD) [120]. It is represented by a type of chromosomal rearrangement that gives cancer cells the capacity to achieve new protein isoforms [121]. As an example, EGFR-KDD is present in human cancers, including glioma, with good response to the targeted therapies against EGFR [122]. Amplification of the EGFR-KDD in the post-treatment biopsy leads to the conclusion that KDD may contribute to the achieved resistance of EGFR TKI, afatinib [122].

### 4.2. EGFR Mutations

Glioblastoma is thought to be a copy number disease, almost 20% of them withholding EGFR point mutations, which are sometimes associated with copy alterations or rearrangements [23]. These point mutations appear more commonly in the extracellular domain [92,97,123] and may be enhanced in EGFR-amplified tumors, but they can also be detected without amplification [23,124]. Clinical detection of the mutations plays an important part in treatment with EGFR inhibitors.

EGFR was one of the first oncogenes to be identified in glioblastoma. In cancer cells, more than 106 EGFR receptors/cell were observed [125]. There are two types of pathological alterations of EGFR in cancers: one is represented by a kinase-activating mutation in EGFR, which can lead to increased tyrosine kinase activity of EGFR and can be primary or secondary to anti-EGFR therapies [126,127], and the second is represented by the overexpression of the EGFR protein and may or may not be associated with gene amplifications [128,129,130,131]. The molecular alterations of EGFR include overexpression, deletion, and amplification. In glioblastoma, the amplification of the EGFR gene is the most frequent RTK mutation (approximately 40%) [132] and is mostly observed in primary GBM. The EGFR amplification is a promoter for invasion, proliferation, and drug resistance [133]. EGFR has also been associated with structural alterations, the most common being EGFR variant III (EGFRvIII). This variant has an in-frame deletion of exons 2–7 from the extracellular domain and is able to transmit growth signaling in a ligand-independent way [134,135]. The expression of EGFRvIII in glioma cells leads to the expression of TGF-α and HB-EGF, resulting in EGFRvIII being involved in generating an autocrine loop with wild-type EGFR expression [119]. Various downstream pathways are activated when EGFR signaling is influenced by genetic alterations. Some of these pathways are represented by phosphatidylinositol 3-kinase (PI3K)/Akt and Ras/Raf/MEK (MAPK kinase)/MAPK (mitogen-activated protein kinase), and they are responsible for cell proliferation, survival, and mobility [136,137]. The tumor suppressor phosphatase and tensin homolog deleted from chromosome 10 (PTEN) negatively regulate the PI3K/Akt pathway. PTEN is emblematic for GBM, being mutated in 20% to 40% of cases [138].

The secondary mutations that contribute to acquired resistance to anti-EGFR therapies are represented by the T790M mutation, accounting for 50% of resistance in non-small cell lung cancer patients that were treated with first- and second-generation TKIs [139,140].

Studies have shown that EGFR overexpression or mutation causes shorter overall survival in patients with GBM [141,142,143].

### 4.3. EGFR Wild-Type

Overexpression of wild-type EGFR is tumorigenic in a variety of types of cells [144,145,146], concluding that EGFR wild-type is oncogenic. The protein levels of EGFR, independent of its phosphorylation status, is strongly associated with the progression of the disease and a bad prognosis of a variety of types of cancers not expressing mutated EGFR [147,148]. Most of the GBMs and low-grade astrocytomas as well as many other types of solid cancers overexpress wild-type EGFR protein [149] throughout the entire tumor. The presence of the overexpression of this protein is mostly associated with negative prognosis [150,151,152,153], being a more common phenomenon than EGFR mutations and promoting disease progression [57]. It was observed by immunohistochemistry (IHC) that almost all cells within the tumor express high levels of EGFR protein.

The reason why cancers expressing/overexpressing wild-type EGFR mutation don’t have a good response to EGFR TKIs is connected to the fact that these types of cancers are not addicted to the EGFR functions for growth and/or survival. However, it seems that TKIs are highly capable to inhibit the growth of wild-type EGFR expressing cells [154,155]. Studies are also showing that cells expressing wild-type EGFR cannot survive after EGFR knockdown by siRNA [156,157,158,159,160]. All this information helps to conclude that although EGFR wild-type cancer cells survival may not be dependent on EGFR’s kinase activity, it may be sustained by EGFR without involving its kinase activity.

### 4.4. EGFR Copy Number Alterations

Focal gene amplification causes an increased number of copies of the gene and is the most frequent EGFR anomaly in adult GBM. This amplification is the result of specific mechanisms, which include the formation of extrachromosomal double minutes (dmins) and the heterogeneously staining regions (HSR) that affect a specific gene locus or chromosomal region. The consequence of this amplification is the increase of the copy number. There is a need to differentiate the terms “gain” that indicates a lower-level increase of gene number caused by different mechanism like tandem gene duplication or polysomy, and the term “amplification” that is used to indicate any level of increase without mechanism implications.

The most common EGFR copy number alterations in glioma is the polysomy of the entire chromosome 7. A substantial percentage of gliomas present other alterations that can lead to a gain of 7q that contains the EGFR gene. Although these chromosomal gains appear frequently in gliomas, they are not GBM specific. More than 40% of the GBM’s present focal EGFR gene amplification having more than 50–100 copies of the gene [23,161]. These amplifications may also appear in association with the co-amplification of other RTK genes [162].

The EGFR copy number alterations can be used clinically for the diagnosis and classification of GBM’s, helping to distinguish them from other tumors with similar histological features, these amplifications being almost specific exclusively to glioblastomas. Although initial studies demonstrated the poor prognosis of the presence of EGFR amplification, later investigations have not confirmed these results [138]. The role of EGFR amplifications as a predictive marker was highly studied in a variety of EGFR inhibitors clinical trials, failing to show good results as an independent biomarker [163,164].

### 4.5. EGFR Rearrangements

#### 4.5.1. EGFRvIII

Almost 30% of the GBMs present the variant EGFRvIII, the most frequent intragenic deletion of EGFR. It appears only in tumors that present with EGFR amplification and presents with intragenic in-frame deletion of exons 2–7, resulting in the overexpression of a truncated receptor that has some missing parts of the ECD. It is not capable of binding ligands, and it is constitutively active due to its ligand-independent dimerization [165]. It is a strong oncogenic promoter, being able to transform cells and generate a more aggressive phenotype in GBMs. The presence of EGFRvIII is specific to GBM, and it can be used as a glioblastoma-specific marker. Although the identification of EGFRvIII is not usually conducted for all patients, it can be performed to identify patients that could be included in specific clinical trials directed against this antigen. The use of IHC helped to identify the heterogeneous extensive expression of EGFRvIII within the tumor, information that was also proved by the patterns observed in genomic assays [166], representing an important hypothesis of the resistance in nonexpressing cells.

#### 4.5.2. Other Deletion Variants

In 71% of the EGFR-amplified tumors, one can identify intragenic deletions. EGFRvII is one of the variants that may be clinically important to the pathogenesis of GBM [161]. Other deletions that were capable of performing cell transformation without ligand were those involving the carboxy-terminal intracellular domain [135,167]. These presented high sensitivity to cetuximab in a xenograft model [168].

### 4.6. EGFR Fusions

When a large RNA-sequencing dataset of glioblastoma was investigated, several in-frame gene fusion and elaborated rearrangements were exposed [169,170]. EGFR was the most common gene fusion in GBM. The most common EGFR fusion was to intron 9 of SEPT14 or to the gene PSPH, leaving the tyrosine kinase domain unharmed [169]. Due to the lacking expression of EGFRvIII in the before-mentioned fusions, the C-terminal deletion of EGFR through gene fusion could be an additional mechanism involved in EGFR activation.

### 4.7. MicroRNAs

MicroRNAs (miRNAs) are endogenous noncoding single-stranded RNAs consisting of approximately 20–22 nucleotides. They regulate gene expression by binding to target protein-encoding mRNA at the posttranscriptional level. MiRNAs have the ability to regulate a large variety of carcinogenic pathways, including the EGFR signaling pathway. MiRNAs have important roles in the initiation, progression, and prognosis of various human illnesses, including cancers. The expression of RTKs can be directly modulated by the activity of microRNAs. These are able to function as both tumor suppressors and as oncogenes [171,172], and they have been proved to play a part in RTK signaling and the regulation of tumor formation. Chromosomal regions that encode oncogenic miRNAs related to the negative regulation of tumor suppressor genes are amplified. Usually, miRNAs that target oncogenes are located at fragile sites where deletions or mutations may appear, resulting in the reduction or loss of expression of miRNAs and the overexpression of targeted oncogenes, thereby initiating cancer development [173].

Recent studies showed that miRNAs participate in the occurrence and development of GBM, regulating numerous biological activities [174,175] such as tumor growth, migration, invasion, and chemotherapy resistance. They may also be used as biomarkers for diagnosis, treatment, and prognosis [176]. Due to their function of regulating the expression of relevant proteins that are involved in the EGFR signaling pathway, they cause tumor cell apoptosis, proliferation, migration, and invasion.

MicroRNA-219-5p can suppress GBM development by silencing EGFR expression, as it is able to bind directly to its 3′-UTR [177]. Alongside others RTKs, EGFR can regulate microRNA-134 in GBM, which acts as a tumor-suppressive center [178]. Further advanced study of microRNAs and RTKs signaling will bring improvements in the cancer therapies domain. It is already known that the association of an inhibitor of microRNA-21 and a monoclonal antibody against EGFR is able to improve the treatment outcome in GBM [179]. MicroRNAs could be an important prognostic marker and help with patient stratification, and further understanding of their role in RTK signaling may help in cancer detection, therapy, and prognosis.

### 4.8. Crosstalk

Many RTKs lead to activation of PI3K/Akt and Rac1 signaling pathways in gliomas, suggesting that they play an important part in downstream signaling for invasion. In his study, Stommel revealed the co-expression of multiple activated RTKs in individual dissociated cells in a primary GBM [180]. Individual cells have the capacity to express a diverse number of RTKs, which in turn have the ability to interact with each other. The phosphorylation of the c-Met receptor as a function of EGFRvIII receptor levels suggests a crosstalk between c-Met and EGFRvIIII signaling [181]. Axl RTK has a similar response [181]. EGFR and EphA2 co-localize on the cell surface and are both expressed in GBM. EphA2 phosphorylation is modulated by EGFR activity. EGFR phosphorylation, downstream signaling, and EGF- induced cell viability are inhibited by EphA2 downregulation [119].

Due to the multiple correlations between RTKs, one single RTK inhibitor is not capable of successfully suppressing the invasion signaling.

## 5. Targeted Therapy—EGFR as Therapeutic Agent

In recent years, many agents, TKIs, and monoclonal antibodies have been developed and used in the clinical treatment of different cancer types with the goal of inhibiting the signaling pathways of EGFR [182,183]. The activity of EGFR can be regulated by binding it to either the extracellular domain or the tyrosine kinase domain. Three generations of TKIs have been developed and approved for use in a variety of cancer types. These TKIs work by targeting signal transduction, binding to the EGFR tyrosine kinase domain, and inhibiting its activity. The first generation of TKIs works by competitive binding with ATP and inhibiting the receptor. As the emergence of acquired resistance to targeted therapies is inevitable [184,185], new agents of second-, third-, and fourth-generation inhibitors and combinations of both TKIs and monoclonal antibodies are now being studied [91,186,187,188,189]. Second-generation TKIs have the ability to irreversibly inhibit all four ERBB receptors, third-generation TKIs were designed to target the T790M resistance mutation that is responsible for about 50% of acquired resistance to the earlier generation of TKIs [190,191,192,193], and fourth-generation TKIs preferentially inhibit the T790M/C797S EGFR mutant that leads to some resistance to the third-generation TKIs [193]. The monoclonal antibodies are receptor blockers and work by preventing EGFR from binding to the ligand by binding to its ECD. The therapeutic effect of anti-EGFR antibodies cannot be solely attributed to the inhibition of the tyrosine kinase activity of EGFR.

The existing data suggest that the tyrosine activity of EGFR is primarily involved in promoting cell proliferation rather than cell survival [194]. Under physiologically relevant conditions, EGFR TKIs and mAbs constantly present anti-proliferative effects [195]. Concerning the impact that TKIs have on cell survival, studies have shown that they are able to induce cytoprotective autophagy, which has the capacity to promote cell survival [196,197].

### The Kinase-Independent Pro-Survival Function of EGFR in Cancer Cells

Studies made in recent years have shown that EGFR has pro-survival functions that are independent of its tyrosine kinase activity in cancer cells. This type of kinase-independent (KID) pro-survival function has been encountered in different types of cancer cells, including several cellular functional domains, such as the plasma membrane, the autophagic machinery, and the mitochondrion. The discovery of this pro-survival function has profound implications for overcoming the challenges of EGFR targeted therapies as it offers an alternative point of view on the clinical failures of already-existing EGFR kinase inhibitors. It may also yield new perspectives on cancers that are expressing/overexpressing EGFR wild-type, as these cancers are innately resistant to EGFR inhibitors. It may be that these cancers cells are more dependent on EGFR’s KID function for survival rather than on its kinase activity for growth.

An explanation of how the acquired TKI resistance develops is that the EGFR’s kinase dependent function is shifted toward its KID pro-survival function, giving cancer cells that are dependent on EGFR kinase activity for growth a way to establish alternative proliferative mechanisms, bypassing the EGFR kinase-dominated pathway that is under constant exposure to TKIs. It has been proven that EGFR can be found in two states in cancer cells: with the capacity to be activated or without this capacity [198]. EGFR with the capacity to be activated behaves according to the canonical mechanisms, while EGFR that cannot be activated is characterized by the inability to autophosphorylate while physically interacting with other proteins at its C-terminal kinase domain [198].

On the basis of existing evidence, it was concluded that the pro-survival function of EGFR is regulated by mechanisms that are not dependent on EGFR kinase function. Targeting the KID pro-survival function of EGFR by lowering its protein level may be helpful in finding more effective ways of targeting EGFR for cancer therapy.

## 6. The tyrosine Kinase Inhibitors

### 6.1. First-Generation EGFR Inhibitors

Gefitinib is the first selective inhibitor of EGFR tyrosine kinase to be made with low molecular weight. Early studies may have shown signs of activity in patients with malignant gliomas [96,163], but in a multicenter phase II study, it showed limited efficacy both as part of the initial treatment regimen and for recurrent disease [96,199]. In a phase I/II trial, when it was combined with radiation for the treatment of newly diagnosed GBM, gefitinib showed no improved overall survival [200].

Erlotinib is an orally active, reversible tyrosine kinase inhibitor (TKI). Studies have not yet demonstrated its efficacy in the treatment of newly diagnosed GBM either used alone or in combination with temsirolimus or bevacizumab [163,164,201,202].

Lapatinib, a first-generation EGFR inhibitor, is a dual TKI that acts as an interrupter on the HER2/neu growth receptor factor. It is being used successfully in breast cancer therapy, but it is not demonstrating results in patients with recurrent GBM [203].

### 6.2. Second-Generation EGFR Inhibitors

Afatinib showed limited efficacy when used as a single agent in a clinical trial in recurrent glioblastoma [204].

Tesevatinib is currently being evaluated for glioblastoma therapy. Tesevatinib had good efficacy results in EGFR-amplified PDX GBM models in vitro but achieved relatively modest performance in vivo despite a high brain-to-plasma ratio [205]. 

Dacomitinib had poor results in a phase II clinical trial for recurrent glioblastoma [206].

Vandetanib was only studied in association with other therapies, either radiotherapy or therapeutic agents, and yielded very poor results.

### 6.3. Third-Generation EGFR Inhibitors

Osimertinib demonstrated an ability to inhibit the constitutive activity of EGFRvIII tyrosine kinase and its downstream signaling in a new study [207].

CM93 is a novel covalent small molecule inhibitor which targets mutant EGFR [208]. In a study performed by Ni et al., this new inhibitor showed promising results expressed as better efficacy in preclinical studies compared with other EGFR-TKIs. It also extended the survival rate of mice with orthotopic GBM allografts [209].

### 6.4. Fourth-Generation EGFR Inhibitors

Fourth-generation tyrosine kinase inhibitors are still under the early phases of preclinical development. The first fourth-generation TKI to be developed is allosteric inhibitor EAI045, which has proven its efficiency in inhibiting the kinase activity of T790M/C797S in combination with Cetuximab but not as a single agent [193].

Recently, the orally bioavailable ERAS-801, a small-molecule EGFR inhibitor, has received an orphan drug designation by the FDA for malignant glioma [210]. In a mouse model, WSD-0922 preferentially inhibited the phosphorylation of several proteins, including EGFR inhibitor resistance proteins [211]. In clinical studies, ERAS-801 demonstrated a survival benefit in 93% of EGFR mutant and/or amplified models and had a higher brain penetrance in addition to prolonged survival compared with other approved EGFR inhibitors such as osimertinib, lapatinib, or erlotinib [210].

## 7. The Monoclonal Antibodies

Nimotuzumab is an antibody targeting the L2 domain of EGFR, a humanized monoclonal antibody responsible for the inhibition of tyrosine kinase activation. The use of nimotuzumab showed relatively favorable results in pediatric diffuse intrinsic pontine glioma [212,213]. It is currently being studied in a phase III trial in concomitant use with radiotherapy and temozolomide in adult GBM patients.

Cetuximab is an antibody targeting the L2 domain of EGFR, and it prevents dimerization and cross-activation, thereby inhibiting downstream signal transduction. It is a chimeric monoclonal IgG1 antibody with high affinity and specificity for EGFR. Studies showed that cetuximab may enhance the cytotoxicity of radiotherapy in patients with EGFR-amplified GBM [214,215]. In 2009, another study showed that there is no correlation between survival and EGFR amplification and the use of cetuximab in patients with recurrent glioblastoma [216]. 

Panitumumab is being investigated in a phase II trial as a GBM diagnostic agent.

bscEGFRvIIIxCD3 is a bispecific T-cell engager antibody (BiTEs) that can bind to the CD3 T-cell coreceptor and recruit cytotoxic T cells. It redirects the T cells towards tumors that express EGFRvIII. Results of the studies made in vitro and in vivo on mice showed that this agent has the ability to destroy GBM cells that express EGFRvIII [217].

mAB806 targets EGFRvIII, its mechanism of action being still unknown [218]. It has the ability to raise the sensitivity of glioma xenotransplants to radiotherapy [219].

## 8. Immunotherapy

CAR T-cells targeting EGFRvIII is a new technology that utilizes a chimeric antigen receptor (CAR) expressed by engineered T cells to recognize their target, in the case of glioblastoma, the target being represented by the EGFRvIII [220]. This technology is still being studied. The results of preclinical studies showed that the use of CAR T-cells targeting EGFRvIII had impressive results in reducing tumor growth [221]. One study showed a lower expression of EGFRvIII in resected tumors from patients treated with CAR T-cell infusion [222]. It has been shown that CAR T cells specific to EGFRvIII have limited efficacy in GBM patients [223].

Regarding the immunologic target, rindopepimut it is a vaccine that uses an EGFRvIII-specific peptide conjugated to a keyhole limpet hemocyanin. The results of a phase III clinical trial showed no improvement in the survival rate of newly diagnosed glioblastoma patients after the use of the vaccine [224,225,226].

T-cell bispecific antibodies (TCB) are engineered molecules that bind both the T-cell receptor and tumor-specific antigens. In a study performed by Iurlaro et al. in patient-derived models expressing EGFRvIII, the EGFRvIII-TCB demonstrated specificity for the mentioned receptor and also promoted tumor cell death together with T-cell activation and cytokine secretion. Furthermore, EGFRvIII-TCB promoted the recruitment of T-cells in the intracranial tumors. It induced tumor regression in humanized orthotopic GBM patient-derived xenograft models [227].

## 9. Targeted Isotopes

^125^I-mAb 425, antibody toxin or radioactive isotope conjugated, is an ^125^I labeled anti-EGFR 425 murine monoclonal antibody developed from mice immunized with A-431 epidermoid carcinoma cells. It binds to the tumor and induces direct cell growth inhibition, complement-dependent cytotoxicity, and activation of the humoral response. A phase II study conducted in 2010 [228] demonstrated that its use was safe and well tolerated, and the reported survival was 15.7 months.

## 10. Nanoparticles

Nanoparticles are vesicular carriers that can protect their composition, resulting in higher bioavailability with a better capacity to penetrate the blood–brain barrier. A high number of agents have been entrapped in a variety of nanoparticles targeting EGFR [229]. Studies are still in progress for cetuximab-conjugated liposomes [230].

## 11. Targeting the Regulation of EGFR Gene Expression

This method is represented by the use of ribozymes, antisense oligonucleotides, siRNA, and miRNA. The post-transcriptional gene expression of RTKs’ signaling pathways is under the control of microRNAs. It was demonstrated that EGFR can be carried by extracellular vesicles (EVs) [231] and that the EVs participating in the cell communication can enhance the intratumoral heterogeneity of glioblastomas [232]. To date, none of the strategies that target the regulation EGFR gene expression have been further studied.

## 12. Challenges to Current Anti-EGFR Therapies

The efficacy of EGFR targeted cancer therapies is obstructed by two major challenges: acquired and innate resistance towards the available drugs.

Drug resistance may be able to explain why the response to EGFR targeted therapies is not efficient for glioblastoma [233]. Two mechanisms were proposed to explain this resistance:The target independence. Not all of the glioblastoma cells express EGFR proteins; therefore, the EGFR inhibitors have no effect on them. A frequent situation encountered in this type of resistance is the loss of extra-chromosomally encoded EGFR. It can appear after the use of small molecule therapies. Small circular fragments of extra-chromosomal DNA act as regulators of dynamic EGFRvIII expression, and they may be involved in the resistance to inhibition. Studies have shown that after the treatment of GBM cells with erlotinib, mutant EGFR was reversibly blocked by producing extra-chromosomal DNA. By seizing the use of erlotinib, the mutations reappeared, resulting in the upregulation of EGFRvIII [234].The target compensation. The GBM cells activate compensatory pathways that are independent of EGFR signaling [183], such as the RAF/MEK/MAPK/ERK-, PI3K/Akt-, and MET-regulated signaling pathways [182,235,236,237,238].

Drug resistance may be compensated for by using multi-target therapies that work to compensate both mechanisms described: targeting truncation mutations for the first mechanism and targeting compensatory proteins for the second mechanism.

Patient mutations that affect the trafficking of therapeutic antibodies are another possible mechanism involved in resistance [239,240].

Innate resistance to anti-EGFR drugs is much more prevalent than acquired resistance. It seems that cancers expressing wild-type EGFR do not respond to TKIs regardless of the expression level of EGFR [241]. It was hypothesized that EGFR is unimportant for those types of cancers that are innately resistant to EGFR kinase inhibitors, but this idea was invalidated by new evidence showing that even in innately resistant cancer types, the downregulation of EGFR proteins causes severe cell death [157]. In other words, even in the cancer cells that are innately resistant to EGFR kinase inhibitors, EGFR is indispensable for survival.

## 13. Conclusions

A better understanding of RTK signaling pathways, getting to know all of the tools they provide us, including their genetic, cellular, biochemical, and structural modeling techniques, can help us to improve patient care. One of the most frequently studied and used RTKs in the battle against cancer is EGFR. There is still a long way to go to understand all of the features of the signaling pathways and their ramifications. The focus must be on the improvement of existent therapy technologies and the development of new ones that may improve the patient’s quality of life.

Focusing on only one type of EGFR-altered tumor (EGFRvIII deletion) may be insufficient in the battle against the complex heterogeneity of GBM [163,242]. Bearing in mind the continuous resistance of GBM to the EGFR inhibition tried so far, we can only hope that the use of new diagnostic methods for assessing EGFR can help us to find new and more-effective methods of inhibition.

## Figures and Tables

**Figure 1 ijms-24-11110-f001:**
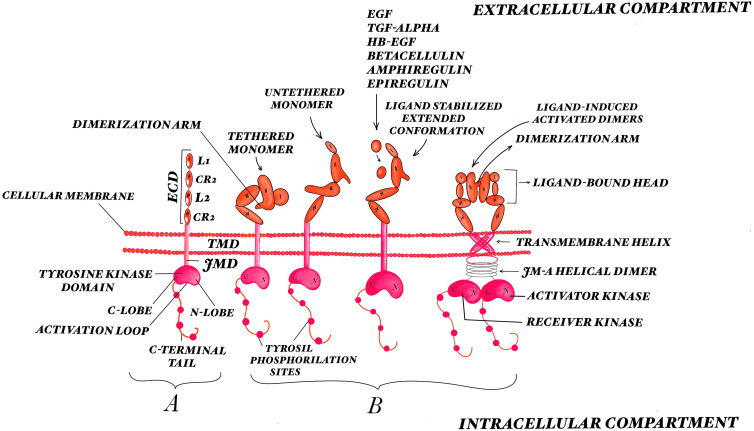
The EGFR structure and mode of activation. (**A**) Overall view of the EGFR structure: the extracellular region incorporates four subdomains: two homologous domains D I and D III, where ligands bind (L1, L2), and D II and D IV, being cysteine rich domains (CR1, CR2); the transmembrane region, the juxtamembrane domain, the tyrosine kinase composed of the C-lobe and N-lobe, and the C-terminal regulatory domains. (**B**) Different steps of EGFR activation: autoinhibitory tethered monomer, untethered monomer, ligand-stabilized extended and ligand-induced activated dimer conformations. EGFR ligands binding to the receptor expose a dimerization motif which determines structural rearrangements that are expressed in the cytoplasmic domain, allowing for the formation of asymmetric dimers between the two juxtaposed catalytic domains. The ligand binding process requires a 130° rotation of Domains I and II about the axis of the Domain II/III junction. In the active state, the residues from Domain IV from the receptor pair form a V-shape. There is an interaction between the carboxy-terminal portion of the activator kinase domain with the amino-terminal portion of the receiver kinase domain. The transmembrane helices of the active receptor interact near their amino-terminal by the extracellular compartment; the inactive receptor transmembrane helices interact near their carboxy-terminal by the intracellular compartment.

## Data Availability

No new data were created or analyzed in this study. Data sharing is not applicable to this article.

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
