# Peer review of "An Overview of EGFR Mechanisms and Their Implications in Targeted Therapies for Glioblastoma"

_ijms, 2023, doi:10.3390/ijms241311110_

Round 1

Reviewer 1 Report

In this review, the authors present an overview of EGFR mechanism and their implications in targeted therapies for glioblastoma. The authors propose that EGFR inhibitor treatment may be a promising therapeutic strategy in glioblastoma. There are additional scientific studies to address to improve the depth of the study and translational potential. Considering the presence of activating EGFR mutations in in 24% of glioblastoma cases and how EGFR mutations play a major role in resistance mechanism, there is an urgent need to address the unmed medical need.

There are some of the studies that can be considered.

1. The authors have not discussed the ongoing study of small molecule EGFR inhibitor ERAS-801 which as given FDA fast track designation status  as a treatment for patients who have glioblastoma with EGFR gene alterations. ERAS-801 has shown significant improvements in brain penetrance and prolonged survival compared with other approved EGFR tyrosine kinase inhibitors like osimertinib, lapatinib, and erlotinib. Erlotinib is a poorly brain penetrant EGFR TKI compared with  ERAS-801.

2. The authors can also discuss blood-brain-barrier (BBB) penetrable EGFR/EGFRvIII inhibitor CM93, which specifically inhibits EGFR mutations, preventing EGFR mutant-mediated signaling and leading to cell death in EGFR mutant-expressing tumor cells.

3. To reach the full potential of EGFR inhibitors, the authors can also include EGFRvIII-targeted immune-based therapy that could potentially treat glioblastoma. A novel EGFRvIII T-Cell Bispecific Antibody has been developed for the treatment of Glioblastoma. EGFRvIII-TCBs can also be considered as a promising therapeutic tool against GBM warranting clinical testing

The review studies documented in this manuscript are very interesting and also can include the following studies above for strengthening the manuscript.

Author Response

Dear Reviewer,

We kindly thank you for your supporting review regarding our article. We took into consideration all your constructive suggestions and we performed the following improvements:

  1. The authors have not discussed the ongoing study of small molecule EGFR inhibitor ERAS-801 which has given FDA fast track designation status as a treatment for patients who have glioblastoma with EGFRgene alterations. ERAS-801 has shown significant improvements in brain penetrance and prolonged survival compared with other approved EGFR tyrosine kinase inhibitors like osimertinib, lapatinib, and erlotinib. Erlotinib is a poorly brain penetrant EGFR TKI compared with ERAS-801.

Answer: We added the comments regarding ERAS-801

  1. The authors can also discuss blood-brain-barrier (BBB) penetrable EGFR/EGFRvIII inhibitor CM93, which specifically inhibits EGFR mutations, preventing EGFR mutant-mediated signaling and leading to cell death in EGFR mutant-expressing tumor cells.

Answer: We added information regarding CM93

  1. To reach the full potential of EGFR inhibitors, the authors can also include EGFRvIII-targeted immune-based therapy that could potentially treat glioblastoma. A novel EGFRvIII T-Cell Bispecific Antibody has been developed for the treatment of Glioblastoma. EGFRvIII-TCBs can also be considered as a promising therapeutic tool against GBM warranting clinical testing

Answer: We discussed the potential therapeutic role of EGFRvIII-TCB in the treatment of glioblastoma.

Best regards,

The Authors

Reviewer 2 Report

The main outcome of this study is to review the EGFR mechanisms and their implications in targeted therapies for glioblastoma. The relationship between glutamine metabolism and immune cell infiltration was then discussed. Processes in which glutamine is involved can influence the development and development of tumors. However, few data have been reported on glutamine metabolism and diffuse glioma. Overall, this is a good study.

 Moderate editing of English language required

Author Response

Dear Reviewer,

We kindly thank you for your review regarding our article. We took into consideration your constructive suggestions and the manuscript has been improved by adding information regarding the involvement of EGFR in development and progression of glioma due to its involvement in the regulation of the proto-oncogene MYC family and in the promotion the glutamine metabolism via ELK1 activation.

Best regards,

The Authors
